# Target State Optimization: Drivability Improvement for Vehicles with Dual Clutch Transmissions

Marius Schmiedt [1,2,*], Ping He [2] and Stephan Rinderknecht [2]

1    Magna PT B.V. & Co. KG, Hermann-Hagenmeyer-Straße 1, 74199 Untergruppenbach, Germany
2    Institute for Mechatronic Systems, Technical University of Darmstadt, Otto-Berndt-Straße 2, 64287 Darmstadt, Germany
*    Correspondence: marius.schmiedt@magna.com

**Abstract:** Vehicles with dual clutch transmissions (DCT) are well known for their comfortable drivability since gear shifts can be performed jerklessly. The ability of blending the torque during gear shifts from one clutch to the other, making the type of automated transmission a perfect alternative to torque converters, which also comes with a higher efficiency. Nevertheless, DCT also have some drawbacks. The actuation of two clutches requires an immense control effort, which is handled in the implementation of a wide range of software functions on the transmission control unit (TCU). These usually contain control parameters, which makes the behavior adaptable to different vehicle and engine platforms. The adaption of these parameters is called calibration, which is usually an iterative time-consuming process. The calibration of the embedded software solutions in control units is a widely known problem in the automotive industry. The calibration of any vehicle subsystem (e.g., engine, transmission, suspension, driver assistance systems for autonomous driving, etc.) requires costly test trips in different ambient conditions. To reduce the calibration effort and the accompanying use of professionals, several approaches to automize the calibration process are proposed. Due to the fact that a solution is desired which can optimize different calibration problems, a generic metaheuristic approach is aimed. Regardless, the scope of the current research is the optimization of the launch behavior for vehicles equipped with DCT since, particularly at low speeds, the transmission behavior must meet the intention of the driver (drivers tend to be more perceptive at low speeds). To clarify the characteristics of the launch, several test subject studies are performed. The influence factors, such as engine sound, maximal acceleration, acceleration build-up (mean jerk), and the reaction time, are taken into account. Their influence on the evaluation of launch with relation to the criteria of sportiness, comfort, and jerkiness, are examined based on the evaluation of the test subject studies. According to the results of the study, reference values for the optimization of the launch behavior are derived. The research contains a study of existing approaches for optimizing driving behavior with metaheuristics (e.g., genetic algorithms, reinforcement learning, etc.). Since the existing approaches have different drawbacks (in scope of the optimization problem) a new approach is proposed, which outperforms existing ones. The approach itself is a hybrid solution of reinforcement learning (RL) and supervised learning (SL) and is applied in a software in the loop environment, and in a test vehicle.

**Keywords:** parameter optimization; deep learning; machine learning; reinforcement learning; driving behavior; dual clutch transmission; launch optimization; launch evaluation

## 1. Introduction

The conference paper "AI-based parameter optimization method applied for vehicles with dual clutch transmissions" [1] was submitted to introduce a new methodology to overcome the problem of a time consuming calibration process [2] of dual clutch transmissions (DCT) by applying the target state optimization (TSO) algorithm. However, can the

TSO algorithm be applied for different vehicle types and still converge faster than common optimization algorithms?

The calibration of the vehicle launch should result in a quick reaction, without acceleration inconsistencies during clutch engagement throughout the entire lifetime of the clutch and under all driving conditions [3]. This is achieved through an embedded software solution, which includes control parameters to influence driving behavior. The ability of DCT is to enable jerkless shifts through its two clutches. This feature comes with the drawback of having a greater calibration effort due to the complex architecture and actuation compared to torque converters [4]. Additionally, new powertrain concepts, like hybrid powertrains, lead to a wider range of engine, transmission, and vehicle combinations, which also result in an increasing calibration effort [4]. The calibration engineer usually has to optimize the calibration parameters of the embedded software in an iterative manner [5]. With new requirements (like legislative requirements), and therefore growing software functions and calibration parameters, the calibration process is getting even more expensive. Striving to optimize the system behavior in a shorter time period has lead to different attempts in research to automize the calibration process. Therefore, the optimization of the launch behavior of vehicles equipped with DCT is investigated in this study.

To automize the optimization, the evaluation of a launch also needs to be defined by transferring subjective feelings into objective measurements [2]. The launch of a vehicle can be evaluated according to its sportiness, comfort, jerkiness, and agility, which are also called evaluation criteria. These are more or less dependent on the following influence factors: engine sound, maximal acceleration, acceleration build-up (mean jerk: first time derivative of the acceleration), and the reaction time [6]. Within a test subject study carried out by He et al. [7], the influence of the maximal acceleration and acceleration build-up is investigated and further transferred to optimization objectives for this study. The study of Skoda et al. [8] investigated the influence of the engine sound on comfort. The original engine sound is reinforced with a sound volume and a sound bass. Both variations lead to poor evaluation results. Kingma [9] investigated the just noticeable difference (JND) of the acceleration and the velocity in longitudinal and lateral directions, and has concluded that the perception threshold regarding acceleration depends on the stimulus profile, while the perception threshold of velocity does not. The perception threshold of the acceleration and jerk is investigated in de Winkel et al. [10] with the help of jerky motion excitation and statistic models. The study by Winkel et al. [10] also investigated JND. According to Weber's law, the JND describes the minimum amount of stimulus intensity that must be changed in order to produce a noticeable variation in sensory experience [11]. However, the JND does not refer to thresholds that result in significantly different evaluations of the launch behavior, because a slight change in auditory and vestibular perception does not necessarily lead to a different evaluation of launch behavior [7]. Haycock [12] investigated the influence of both acceleration and acceleration build-up on the motion strength, but the researched stimulus for the acceleration is only up to $1 \, m/s^2$, and for the acceleration build-up up to $3 \, m/s^3$. These values are smaller than the reachable value in a vehicle launch. In the study by He et al. [7], the evaluation difference threshold (EDT) is proposed. The EDT describes how much the stimulus intensity must be changed, in order to generate a variation of an evaluation (the significantly varying evaluation is determined within statistic tests). The EDT is generally larger than the JND. There are hardly any studies investigating the EDT for the launch behavior for determining the variation of an evaluation according to sportiness, comfort, and jerkiness. The identified EDTs of this study offer an orientation for choosing reference values for the automated calibration of the launch behavior with different optimization algorithms.

Genetic algorithms (GA) are a common approach to solve multi-objective optimization problems like the calibration of the parameters of the transmission control unit (TCU).

In general, GA are inspired by genetic evolution in nature and usually an initial population is determined randomly [13]. Each population consists of several individuals. For each individual, a fitness value is assigned based on the behavior the individual



performs in the desired environment (how it fits to the optimization objectives) [14]. GA also typically consists of the steps: selection, mutation (spontaneous altered genetic [15]), and recombination (inheriting the shared genetic of parents to a child [16]). Based on the fitness value, the individuals are selected for mutation and recombination [17]. After mutation and recombination, new individuals are part of the following generation. In using GA, the population size and the number of generations have to be defined [13].

The GA NSGA-II algorithm from Deb et al. [13] is a well-tested algorithm for solving such kinds of multi-objective optimization problems, and has been applied in a wide range of optimization problems [2,18–22].

Kahlbau [20,21] used this algorithm in the scope of parameter optimization for optimal shift control of dual clutch transmissions. The shift control has been optimized through the minimization of the jerk, and the minimization of the derivative of the jerk, as optimization objectives. Additionally, the optimization of the launch process with the same objectives as Kahlbau [20,21] have been applied by Wehbi et al. [2] for DCT. Wehbi et al. [2] additionally introduced another optimization objective by measuring the time from start of the launch (acceleration pedal unequal zero) to synchronization (clutch slip speed equal zero). Bachinger et al. [23] optimized the vehicle launch using the differential evolution algorithm for vehicles with DCT. For optimization, the clutch closing time and a scaling factor for clutch slip speed have been applied as objectives. GA have also been applied in other fields, like the optimization of calibration parameters of the engine control unit (ECU). To optimize ECU calibration parameters, a novel GA has been invented based on existing GA. With a special design of experiments, an initial population is identified for reducing the evaluation time. For this aim, a gray-box model based on a neural network is used to model the engine [24]. Additionally, Huang [25] and Zhong et al. [26] applied GA to optimize driving behavior by improving the clutch engagement of a transmission.

GA, however, have some disadvantages, such as the long computation time [27] and the potential of remaining in a local minimum [28]. The parameter population size (number of individuals per generation), for example, directly influences the computation time and affects the quality of the evaluation [29]. The quality of the solution of a GA is also influenced by choosing the right parameters for crossover and mutation [30].

As deep learning has gained attention since mastering challenging tasks like speech recognition and object recognition [31], reinforcement learning (RL) has also evolved. In RL, information about the problem is obtained by the interaction of an agent with an environment [32], such as the famous example of a Q-learning algorithm playing Atari Games [33]. The RL algorithm called agent is a computational approach mimicking the learning behavior of creatures by making machines interact with an environment. The agent takes actions to influence the environment and the environment responds with a state it takes in after receiving the action. The state is fed back to the agent on which basis the next action is chosen to influence the environment towards a defined goal (which is related to the state). The quality of an action is measured with the reward. With the reward, the agent learns which actions are preferably to take. An action which leads to an environmental state towards an optimization goal is rewarded higher than a contrary one. Therefore, the aim is to increase the reward with actions of higher quality [32].

Usually, RL algorithms are used to solve combinatorial problems like the traveling salesman problem [34,35]. Nevertheless, the improvement of clutch engagement with RL has also been in the scope of research. Xiaohui et al. [36] tried to minimize the jerk and the friction losses for clutch engagement during launch within simulations by varying the parameters of a PID controller with a RL algorithm. Another approach was the minimization of the piston velocity and the engagement time of the clutch by Gagliolo et al. [37] and Van Vaerenbergh et al. [38] applying a RL algorithm. Their approach led to promising results after a few hundred epochs. The optimization problem of Brys et al. [39] is similar to the work of Gagliolo et al. [37] and Van Vaerenbergh et al. [38]. Within a simulation, a RL algorithm is applied. The study focuses on multi-objective aspects and on the effects of scalarization on the performance of the algorithm. The optimization of the quality of

the clutch engagement with a RL algorithm, while ensuring an immediate response, is investigated by Lampe et al. [40].

The optimization of the calibration parameters has not been part of the studies which focused on clutch engagement. One possible reason is that the calibration problem is not a combinatorial one; another could be the fact that RL needs many optimization epochs to converge, as illustrated in [37,41].

The drawbacks of RL and GA approaches in the optimization of the launch behavior through optimizing calibration parameters makes these approaches difficult to use in the regular development process of dual clutch transmissions. To overcome these issues, the TSO algorithm by Schmiedt et al. [1] is applied for the calibration problem. The TSO algorithm is also based on machine learning (ML) as a hybrid approach of RL and supervised learning (SL). The application of the TSO algorithm increases the efficiency of the calibration process within different test environments.

This paper aims to dive deeper in the TSO algorithm and illustrates the influence of the activation functions of the used neural networks on optimization behavior. Further, it illustrates a significant advantage compared to previous approaches applied for such optimization problems, but also mentions the drawbacks of the algorithm.

The optimization problem and the results of the subject test studies are described in Section 2. In Section 3 the results of existing benchmark algorithms are illustrated. The TSO algorithm is explained in Section 4. The results are outlined in Section 5. Section 6 discusses and interprets the results and finally, in Section 7, the conclusion is presented.

## 2. Optimization Problem and Objective Functions

The behavior of a vehicle and its components can usually be influenced through calibration parameters. These parameters are part of the software of the corresponding vehicle component, which offers the possibility of adjusting the software to different types of vehicles without changing software functions, or the entire software. These parameters also enable the possibility to influence the system behavior to meet customer requirements. The calibration parameters have to be set iteratively under different environmental conditions to ensure a functioning vehicle with the most achievable quality and by meeting any requirement in most possible conditions. Therefore, many calibration parameters are required within the software, which results in a high calibration effort. To influence the driving behavior of a vehicle these calibration parameters are implemented in the TCU software. To set these parameters, experienced engineers usually optimize the driving behavior based on their subjective feelings. Therefore, the assessment of the driving behavior can vary between different engineers, due to the fact that the evaluation is biased by personal preferences. This obviously can lead to further adjustment loops and an increased calibration effort [6]. Hence, to reduce the workload of an engineer, the automated optimization of the drivability is strived to be deployed in the development process. Therefore, subjective feelings have to be transferred into objective measurements to let the optimization algorithms monitor the quality of the driving behavior without the input of an engineer [2]. The transfer of these subjective feelings in this study takes place by measuring the launch behavior with four different objectives. The objectives are: (1) the acceleration objective, (2) the reaction time objective, (3) the engine speed objective, and (4) the clutch torque objective.

Before applying these objectives in the automated optimization, the objectives are first investigated in a driving simulator. The driving simulator allows the investigation of the influence factors in a reproducible test environment and the conduct of the test subject studies. With the help of statistical tests, the EDTs can be detected, which provide certain optimization targets for the calibration of the powertrain.

The driving simulator is developed for the investigation of human perception during longitudinal drive maneuvers and has the ability to simulate the dynamics in the longitudinal direction of the vehicle. It is able to represent the acceleration and deceleration by combining translatory and rotatory movements of the driver cabin [42]. By applying

virtual reality technology, simulating the driving environment (e.g., the streets, traffic signs, landscape), and synthesizing the engine sound the test subjects face a reality-close driving situation. Besides the visual and auditory senses, the simulation of the haptic sense is considered as well. More details about the driving simulator can be found in [43].

In the studies, the test subjects experienced launch procedures, with varying objectives (engine speed with drops, maximal acceleration, jerk, and reaction time). Further, the subjects had to evaluate these launches on a scale from 1 to 5 according to the evaluation criteria: comfortable, acceptable, jerky, sporty, and agile. The evaluations were processed and analyzed with statistical tests. The utilized tests in the studies were the variance analysis [44], the *t*-test, and the Wilcoxon-test. These methods study the influence of the objectives and can detect whether the evaluations of launches are significant differences when the objectives vary. He et al. [7] introduced the test subject study and the study results of the objectives of maximal acceleration and jerk in detail. The investigation process of other objectives is comparable.

The optimization objectives are separated into customer objectives (Section 2.1), as well as in discomfort objectives (Section 2.2).

### 2.1. Customer Objectives

The customer objectives are based on customer requirements and hence a measured value ($v_{meas}$) is compared to a customer defined reference value ($v_{ref}$). The absolute deviation of these values is normalized so the objective values range between zero and one:

$$obj_{customer} = \left| \frac{v_{meas} - v_{ref}}{v_{ref}} \right| \tag{1}$$

The normalization is introduced to weigh the objectives equally and to make them comparable and hence enables the possibility to set a relative tolerance. Therefore, if such an objective value is lower than 0.1 it is set to zero. Hence, a tolerance of 10% based on the reference value is introduced:

$$v_{ref} \cdot 0.9 \leq v_{meas} \leq v_{ref} \cdot 1.1 \tag{2}$$

The customer objectives of Sections 2.1.1 and 2.1.2 is therefore finally determined with the following equation:

$$obj_{customer,Tolerance} = \begin{cases} 0 & v_{ref} \cdot 0.9 \leq v_{meas} \leq v_{ref} \cdot 1.1 \\ \left| \frac{v_{meas} - v_{ref}}{v_{ref}} \right| & otherwise \end{cases} \tag{3}$$

#### 2.1.1. Acceleration Peak and Acceleration Build-Up Objective

Humans cannot perceive speed but can perceive acceleration and jerk [45]. The perception of the driver is therefore influenced by the acceleration and the acceleration build-up during launch, since the aim of a vehicle launch is to increase speed from standstill. The maximal acceleration and the acceleration build-up are varied in the test subject study by He et al. [7] to investigate the influence on the evaluation criteria: sportiness, comfort, and jerkiness. An example of an acceleration profile is shown in Figure 1.

The test subject study of He et al. [7] is introduced to identify the EDTs of the acceleration for sportiness and comfort. The EDT for sportiness is depending on the intensity of the maximal acceleration. It is lower than $0.5 \, \text{m/s}^2$ when the maximal acceleration is lower than $3 \, \text{m/s}^2$. The EDT is between $0.5 \, \text{m/s}^2$ and $1 \, \text{m/s}^2$ as the maximal acceleration continues to rise.

The EDT for comfort is between $0.5 \, \text{m/s}^2$ and $1 \, \text{m/s}^2$ for all the tested maximal accelerations. It is observable that the EDT for sportiness is more sensitive to the stimulus intensity than the EDT for comfort due to the fact that the EDT for sportiness tends to be lower than the EDT for comfort (for lower maximal accelerations). To determine the

concrete EDTs, more test subject studies need to be conducted. However, the maximal acceleration does not influence the evaluation of jerkiness significantly.

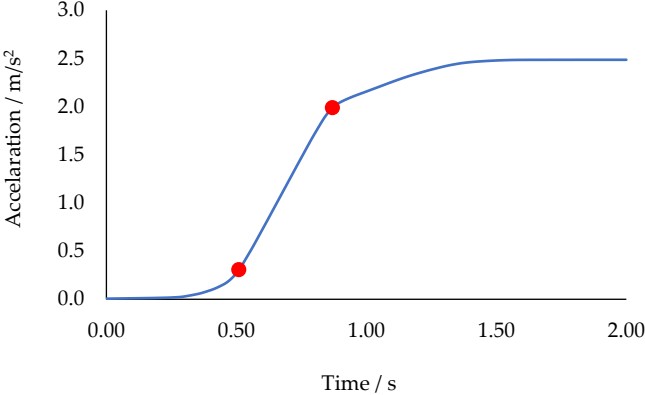

**Figure 1.** Acceleration profile of a launch [7].

In the test subject study by He et al. [7], the acceleration build-up refers only to the mean jerk during the rise of the acceleration. The mean jerk is determined starting from 15% and ending at 85% of the maximal acceleration [6], so that the transitions from standstill to a steady growing acceleration, and further the transition to the maximal acceleration, are excluded from the calculation of the mean jerk. The starting and ending points are marked in Figure 1 in red. The results indicate a logarithmically changing EDT of the acceleration build-up, which matches the Weber–Fechner law [11]. The EDTs of the acceleration build-up for the criteria of sportiness, comfort, and jerkiness, are greater than 2 m/s$^3$, while the acceleration build-up is smaller than 7 m/s$^3$. As the acceleration build-up increases, the thresholds for the three criteria become smaller than 2 m/s$^3$. The influence of the acceleration build-up for jerkiness and comfort are contrary. The higher the maximal acceleration and the acceleration build-up, the more jerky and the less comfortable the launches are [7].

Within the study by He et al. [7] a regression analysis was carried out to identify the borderline between negative and positive evaluations. The borderline represents a combination of the maximal acceleration and the acceleration build-up. Negative launch behaviors (e.g., discomfort) are marked in blue. It is observable that for the borderline the maximal acceleration and the acceleration build-up influence each other mutually for any of the three criteria. When, for example, the acceleration is increasing the acceleration, build-up is decreasing. The results are shown in Figure 2.

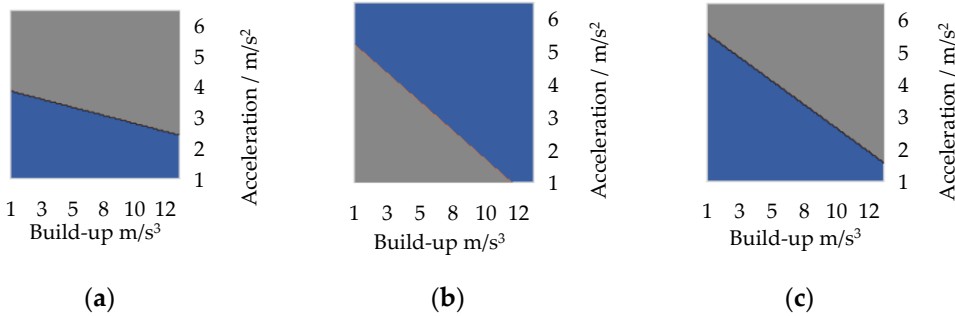

|  |  |  |
|:---:|:---:|:---:|
| (a) | (b) | (c) |

**Figure 2.** Borderlines for the criteria of: (**a**) sportiness, (**b**) comfort, and (**c**) jerkiness, the blue fields represent the negative evaluation areas [7].

The demands for a sporty and a comfortable launch are conflicting [20]. Minimizing the maximal acceleration and the acceleration build-up is improving the passengers comfort [46]. High accelerations and mean jerks (shorter build-up times) in contrast tend to be sportier [2]. Obviously, to increase the speed of a vehicle the maximal acceleration and further the mean jerk must be greater than zero, but they also should not be too high

either. Therefore, a reference value with regard on the subject study should be chosen for the maximal acceleration ($a_{max,\ ref}$) and the acceleration build-up ($\dot{a}_{build,ref}$), respectively, which are compared with the measured values $a_{max}$ and $\dot{a}_{build}$. So, the maximal acceleration objective is determined with the following equation:

$$obj_{Max,Acc} = \begin{cases} 0 & a_{max,\ ref} \cdot 0.9 \le a_{max} \le a_{max,\ ref} \cdot 1.1 \\ \left| \frac{a_{max} - a_{max,\ ref}}{a_{max,\ ref}} \right| & otherwise \end{cases} \tag{4}$$

Accordingly, the acceleration build-up objective is determined with the equation below:

$$obj_{Build,Acc} = \begin{cases} 0 & \dot{a}_{build,ref} \cdot 0.9 \le \dot{a}_{build} \le \dot{a}_{build,ref} \cdot 1.1 \\ \left| \frac{\dot{a}_{build} - \dot{a}_{build,ref}}{\dot{a}_{build,ref}} \right| & otherwise \end{cases} \tag{5}$$

As mentioned in Section 2.1 the objectives are zero if the measured values are within a range of 10% compared to the corresponding reference values. Both objectives are customer objectives since they are influenced by customer requirements but preferably should be chosen with respect to the study by He et al. [7].

### 2.1.2. Reaction Time Objective

The reaction time measures the time the driver has to wait from acceleration pedal actuation to a noticeable acceleration, which is according to Simon [6] if 15% of the maximum acceleration is reached.

To identify an optimal reaction time another subject study is introduced. The study aims to find the influence of the reaction time on the following criteria: sportiness, comfort, and agility. The study is carried out in the same procedure as the prior ones. The reaction time is varied between three values: 250, 550, and 850 ms. Additionally, the accelerator pedal position is varied between 20%, 40%, and 60%, during these tests (the driver expects different vehicle reactions under different accelerator pedal positions). The reaction time has a significant influence on the criteria sportiness and agility, but little influence on the comfort criterion. For the sportiness and the agility, the EDT of the reaction time depends on the accelerator pedal position. The EDT for sportiness is greater than 300 ms when the acceleration pedal position is 20%, which refers to a slow launch expectation. With a greater accelerator pedal position, the drivers are more sensitive to reaction times smaller than 550 ms. The EDT of the reaction time for sportiness is smaller than 300 ms when the reaction time is lower than 550 ms. However, the test subjects cannot differ reaction times between 550 ms and 850 ms, this means the EDT is greater than 300 ms when the reaction time is greater than 550 ms. In contrast, the influence on the agility is more sensitive. In launch situations with 20% accelerator pedal, the EDT for agility is greater than 300 ms when the reaction time is smaller than 550 ms. However, it is smaller than 300 ms when the accelerator pedal position is greater than 20%.

The results of this subject study can be used for setting the reference value (depending on the accelerator pedal position) which is compared with the measured reaction time:

$$obj_{ReactionTime} = \begin{cases} 0 & t_{res,ref} \cdot 0.9 \le t_{res,\ meas} \le t_{res,ref} \cdot 1.1 \\ \left| \frac{t_{res,\ meas} - t_{res,ref}}{t_{res,ref}} \right| & otherwise \end{cases} \tag{6}$$

Analogue to the acceleration and acceleration build-up objective of Section 2.1.1 the reaction time objective is a customer objective and hence the reference value should be greater than zero and again chosen with respect to the study by He et al. [7].

### 2.2. Discomfort Objectives

In contrast to the customer objectives of Section 2.1 the discomfort objectives are not dependent of a reference value. Instead, the objectives shall be zero for achieving an optimal

result since any value greater than zero has a negative impact on the driving behavior only. The discomfort objectives are illustrated in Sections 2.2.1 and 2.2.2.

2.2.1. Engine Speed Objective

The engine speed behavior is affecting the perception of the driver during the launch, through influencing the acoustics of the vehicle. Therefore, a dropping engine speed (Figure 3b) directly affects the evaluation of the driving comfort. With the test subject study, the EDT is evaluated by varying the engine speed drops between 0, 250, 500, and 750 rpm. The test subjects evaluate every variant according to the criteria: comfortable and acceptable. In this study, it is observed that the engine speed drop leads to a significantly different evaluation when the speed drops more than 250 rpm which hence is the EDT. Therefore, the EDT for both criteria (comfort and acceptance) are equal.

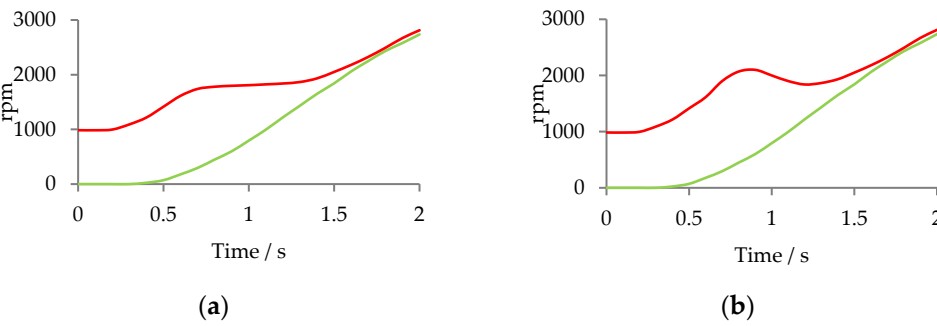

**Figure 3.** (**a**) Engine speed (red) does not drop, (**b**) engine speed drops; Input-shaft speed is green.

The launch behavior, therefore, can be improved by ensuring a solely rising engine speed [47] or should at least not drop more than 250 rpm according to the subject study. Additionally, during clutch slip phases the engine speed can be influenced by the clutch torque [5] and hence decreasing the engine speed is only achieved by increasing the clutch torque over the engine torque, or by requesting an engine intervention. Both options to decrease the engine speed should be avoided since increasing the clutch torque can result in discomfort [25] and requesting an engine intervention can result in increased pollutant emissions [48]. Therefore, it is mandatory to strive for a solely increasing engine speed. These requirements are transferred into the following optimization objective:

$$obj_{EngSpd} = \frac{n_{drop}}{1000 \; rpm} \tag{7}$$

The objective is normalized to keep the objective value within a similar range as the customer objectives (see Section 2.1—between zero and one). Due to the fact that He et al. [7] investigated an engine speed drop of less than 250 rpm as not critical, a tolerance is also implemented for this objective. During evaluation a stricter threshold of 100 rpm is desired and thus, the objective is normalized with 1000 rpm so that comparably with the customer objectives a calculated objective value lower than 0.1 results in an objective value of zero.

$$obj_{EngSpd} = \begin{cases} 0 & \frac{n_{drop}}{1000 \; rpm} \leq 0.1 \\ \frac{n_{drop}}{1000 \; rpm} & otherwise \end{cases} \tag{8}$$

Figure 3 illustrates the behavior of the engine speed. The left measurement (a) is measured with a vehicle, the right one (b) is modified manually to illustrate the behavior.

2.2.2. Clutch Torque/Jerk Objective

Compared to acceleration, humans are even more sensitive to high jerks (first derivative of the acceleration). High accelerations are related to discomfort, but passenger comfort is also tied to the change of acceleration of a vehicle [49]. Müller et al. [50] detected the JND

of the jerk with an experiment in a vehicle. It amounts to 0.53 m/s$^3$ with a 95% confidence interval from 0.07 m/s$^3$ to 1 m/s$^3$.

The longitudinal acceleration of the vehicle is directly linked to the clutch torque ($tq_{clu}$) since the clutch torque is directly transmitted to the wheels during slip phases. Therefore, it is necessary to take care of the clutch torque gradient during the control of the clutch since during clutch engagement every inconsistency is related to discomfort [25]. The objective analyzes the clutch torque during launch for local minima, which indicates an inconsistency in the acceleration signal during launch. A local minimum is found if the first derivative of the clutch torque is zero and the second derivative of the clutch torque is greater than zero. The number of local minima is counted and optimally has to be zero to avoid changes of jerk:

$$obj_{Torque} = \sum_{i=1}^{n} \left[ \dot{tq}_{clu}(t) = 0 \ \wedge \ \ddot{tq}_{clu}(t) > 0 \right] \tag{9}$$

In contrast to the other objectives the raw value is not normalized, so if an inconsistency occurs the objective value is equal or greater one and eliminates the chance of having a successful launch in case of an objective value different to zero.

Although the equation above is theoretically correct, an application is difficult since the measured clutch torque in a real vehicle is a noisy signal with several local minima. Therefore, the signal is filtered with a one dimensional gaussian filter [51] for reducing the noise. Despite filtering, examining local minima numerically is still difficult since the measured signal is sample based and finding an exact zero of the derivatives is often not possible. Instead, the derivative of the filtered torque is investigated with the following Algorithm 1.

---

**Algorithm 1.** Local minima detection of the clutch torque.

**Begin**

    **Set** tfg = gradient(torque_filtered)
    **Declare** osc
    **Set** q = 0
    **Set** i = 1
    **While** i < length(tfg)
        **if** (tfg [i] < 0 && tfg[i − 1] ≥ 0) | | (tfg [i] ≥ 0 && tfg[i − 1] < 0) **then**
            **Set** osc[q] = i
            **Set** q++
        **end**
    **end**
    **Set** l = 0
    **Declare** lp
    **For Each** o **in** osc
        **if** torque_filtered[o − 1] > torque_filtered[o] **then**
            **Set** lp[l] = o
            **Set** l++
        **end**
    **end**
    **Set** $obj_{Torque}$ = length(lp)
    **Return** $obj_{Torque}$

**End**

---

The derivative of the filtered torque is observed for changes of the sign. If a change of the sign is detected (*osc*) either a maximum or minimum is detected. Further, the filtered torque is investigated with the indices of the sign changes. If the value of the found extrema is lower than the value of the former found extrema it is an indication of a local minima. The number of local minima is the value of the objective. The influence of the filtering is illustrated in Figure 4.

The number of local minima (determined with Algorithm 1) for the unfiltered signal of the test vehicle on the left (Figure 4a) is 25 since the signal has a noisy behavior. The result indicates an uncomfortable driving behavior. Nevertheless, such a behavior with small amplitudes is not noticeable and hence does not affect the assessment negative. In contrast, Algorithm 1 applied on the filtered signal (Figure 4b), indicates a driving behavior without discomfort since no local minima is determined. A typical behavior of a discomfort causing clutch torque is illustrated in Figure 5.

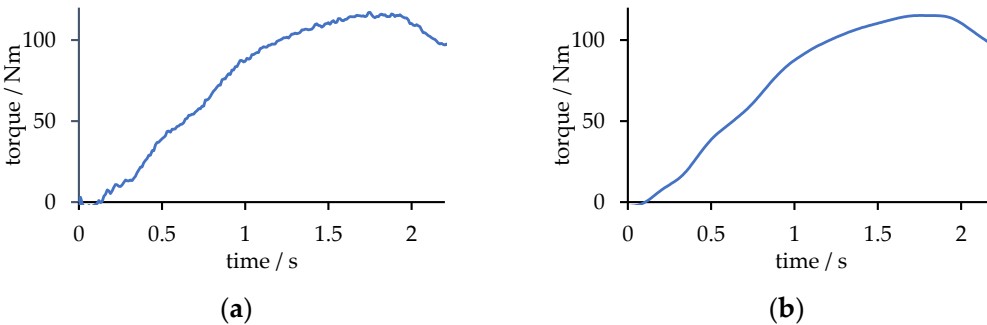

**Figure 4.** Influence of the filtering on the signal (**a**) unfiltered, (**b**) filtered.

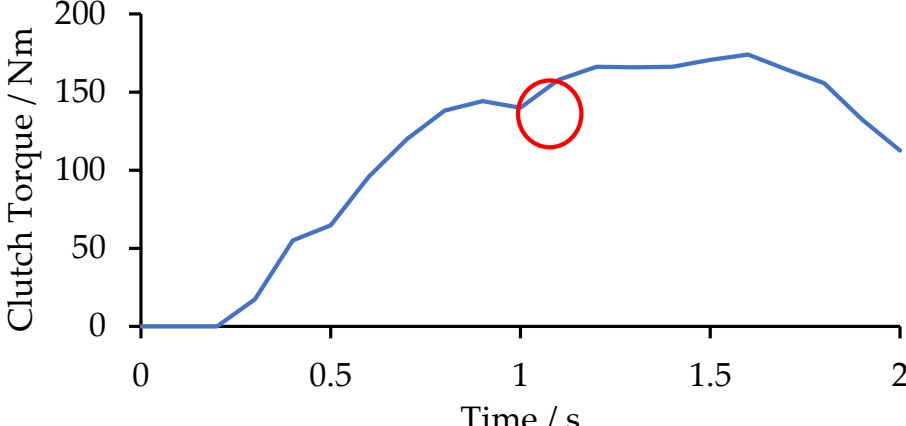

**Figure 5.** Clutch torque with one local minimum.

A local minimum of the clutch torque is indicated with the red marker which would be noticeable as discomfort.

*2.3. The Reward*

As mentioned prior, to measure the quality of an action the reward in RL algorithms is introduced. The reward is based on the environmental state and is strived to be maximized by the algorithm. For this problem, the reward is defined as the sum of the objectives, but negated. The greater the reward the better and vice versa. If the reward is zero, then every objective is zero as well and hence the state equals the target state which indicates a successful launch. The reward for determining the quality of a calibration within the software in the loop environment is determined as follows:

$$reward = -\left( obj_{EngSpd} + obj_{Max,Acc} + obj_{Build,Acc} + obj_{ReactionTime} \right) \tag{10}$$

The target state for any optimization within the software in the loop environment has been set accordingly to Table 1 with respect to the study by He et al. [7]:

**Table 1.** Target state for the software in the loop environment.

| Objective | Value | Unit |
|---|---|---|
| Engine Speed Drop | 0.00 | rpm |
| Acceleration Peak | 3.25 | $m/s^2$ |
| Acceleration Build-up | 6.50 | $m/s^3$ |
| Reaction Time | 0.50 | s |

The state has been reproducibly reachable within simulations and has been used for testing every benchmark optimization algorithm of Section 3.

In the further progress of the study, the application of the algorithms within a test vehicle were at first also carried out with the same target state of Table 1. It turned out that from a practical point of view, the launches with an acceleration build-up different from the reference value could have also felt subjectively satisfying, but also some launches meeting the requirement felt bumpy. Therefore, instead of meeting the reference value of the acceleration build-up, the discomfort objective of Section 2.2.2 has been introduced. It measures if an inconsistency of the clutch torque, and hence the acceleration, is noticeable to avoid bumpy launches. The reward function thus, changes in the test vehicle to:

$$reward = -\left(obj_{EngSpd} + obj_{Max,Acc} + obj_{Torque} + obj_{ReactionTime}\right) \tag{11}$$

Further, the reward is also used to measure the quality of the genetic algorithm by summing up the objectives.

### 2.4. Software in the Loop Environment

As mentioned in Section 2.3, the different algorithms are tested within a software in the loop environment. The benefit of testing optimization algorithms in a virtual vehicle lies in the fact of having reproducible results without influences of ambient conditions. The software in the loop environment is set up with the tool Silver a product of Synopsys. Silver is a virtual ECU platform and enables the development of some tasks without the use of hardware [52]. Within Silver, the entire powertrain model is implemented. The internal combustion engine model thereby acts as a torque source while keeping auxiliary torques into account. Additionally, the hardware models of the vehicle itself for determining the driving resistances as well as the transmission model with its inertias are implemented to closely approach as a digital twin of a real vehicle. Due to confidential reasons the models cannot be illustrated in detail. The control software of the TCU is the same as the one in a real vehicle and hence software functions can be tested without the use of expensive prototypes. Since the system behavior of the powertrain model is not equal to a real vehicle, the software in the loop environment is not well suited for calibration tasks. However, the different calibration techniques to automize the calibration process can be tested well in the software in the loop environment, due to its ability of providing reproducible results. The most promising algorithms are further tested in a test vehicle the results are illustrated in Section 5. The tests are all performed/simulated with zero slope, at sea level, and at 20 °C.

### 2.5. Optimization Parameters

During clutch slip phases the gradient of the engine speed is dependent on the difference between the engine torque and the clutch torque. If the clutch torque e.g., is zero and the engine torque is greater than zero, the engine speed is increasing since there is no resistance from the road applied. Generally, if the engine torque is greater than the clutch torque the engine speed is increasing. Accordingly, if the clutch torque is equal to the engine torque the engine speed is remains constant. The engine speed is decreasing if the clutch torque is greater than the engine torque during clutch slip phases. Thus, the engine speed can be controlled by the clutch. Therefore to calibrate the clutch engagement

the engine target speed is shaped [5]. The launch itself is divided into different phases [53]. The three phases are illustrated in Figure 6.

1. The engine speed is increasing, vehicle accelerating (red).
2. The engine speed is steady until the input-shaft speed is almost equal to the engine speed (blue).
3. The engine speed and its speed gradient are adjusted to the input-shaft speed and its speed gradient (for a smooth clutch engagement (yellow)).

The objectives of Sections 2.1 and 2.2 are optimized with five calibration parameters. These parameters are dependent on the driver request torque (derived from the accelerator pedal position):

- Parameter 1—phase 1: percentage of the engine torque which should be used to accelerate the engine speed to phase 2 (low value: quick vehicle acceleration response but the engine speed increases slowly—sluggish vehicle acceleration, high value: fast increase of the engine speed worsened vehicle acceleration response).
- Parameter 2—phase 1: minimum value of the engine torque which should be used to accelerate the engine to phase 2 (active if the value of parameter 1 is to low).
- Parameter 3: P-gain control value to control the behavior of the engine speed regarding the engine target speed.
- Parameter 4—phase 3: time for reducing the slip speed in phase 3.
- Parameter 5—phase 3: engine speed which should be reached at the end of phase 3.

As mentioned prior, these parameters are dependent on different signals (e.g., driver request torque). An example of such a map is illustrated in Table 2:

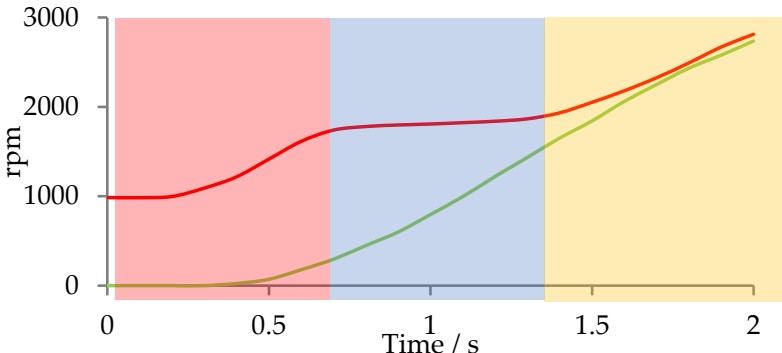

**Figure 6.** Three phases of the launch.

**Table 2.** Example of a calibration map—Parameter 3: P-gain.

| P-gain | | Engine Speed Error/Rpm | | | | |
|---|---|---|---|---|---|---|
| | | −500 | −250 | 0 | 250 | 500 |
| Driver request/Nm | 0 | $z_{11}$ | $z_{12}$ | $z_{13}$ | $z_{14}$ | $z_{15}$ |
| | 150 | $\mathbf{z_{21}}$ | $\mathbf{z_{22}}$ | $\mathbf{z_{23}}$ | $\mathbf{z_{24}}$ | $\mathbf{z_{25}}$ |
| | 300 | $z_{31}$ | $z_{32}$ | $z_{33}$ | $z_{34}$ | $z_{35}$ |

The engine speed error is the deviation of the engine target speed and the measured engine speed. Table 2 illustrates that the value of the P-gain is varied for different engine speed errors (between the nodes the values are interpolated).

Since not every value in the map is required to be changed to optimize the desired driving maneuver (if the driver request is 150 Nm the nodes of 0 Nm and 300 Nm does not affect the behavior), only the ones which affect the behavior are adjusted. This is exemplary illustrated with the marked row of Table 2. For optimizing the desired driving

maneuver only, the marked row needs to be adjusted. Therefore, from these five calibration parameters, 22 values are derived which have to be adjusted during optimization.

## 3. Benchmark: Self-Learning Algorithms

To solve the optimization problem of Section 2, several self-learning algorithms have been applied to the problem. For comparing these algorithms, the virtual vehicle of Section 2.4 is used to ensure reproducible results. The behavior of the vehicle is therefore only influenced by varying parameters of the TCU (Section 2.5) to calibrate the launch behavior. These parameters are set with the following algorithms: the well-known GA NSGA-II and the RL approaches: Deep Deterministic Policy Gradient (DDPG) [54], Proximal Policy Optimization (PPO) [55], Advantage Actor-Critic (A2C) [56] and Soft Actor-Critic (SAC) [57].

To validate the algorithms, vehicle launches are performed within the software in the loop environment and evaluated regarding the optimization objectives of Sections 2.1 and 2.2. Each algorithm is tested within five test runs to neglect the influence of having favorable or unfavorable starting conditions. The starting conditions can vary since, usually in optimization algorithms, the first iterations are performed with a set of randomly chosen parameters. Zaglauer [24] introduced for this purpose a design of experiments to identify a starting population, which gains a wide spread of results within the search space and, therefore, gathering the most possible information out of one iteration. The benefit of having a greater knowledge about the search space is that the evaluation time is shortened [24]. The drawbacks of the method are that domain knowledge is required to set up such a design of experiments and the method is difficult to generalize. Therefore, for this study, such a design of experiments is not intended, instead, the aim is to identify methods which generalize for different optimization problems. Thus, the five test runs are introduced to compare the algorithms while minimizing the influence of starting conditions.

A single test run contains 1000 consecutive iterations (1000 vehicle launches within the software in the loop environment). The number of successful iterations (as explained in Section 2.3 a launch is successful if the reward is zero) and the required iterations until the first successful iteration occurs is determined in each test run. The optimization usually stops if the first successful result is observed in a test vehicle. During these tests the optimization continues until 1000 iterations are reached to ensure a sustainable quality of the results and to eliminate the chance of having a random hit. To eliminate outliers the results of these five test runs are averaged. The results of the five test runs are shown in Table 3. The column $\overline{x}_S$ is the average of the successful iterations, $\sigma_S$ denotes to the standard deviation and the columns $min_S$ and $max_S$ indicates the least respectively the highest number of successful iterations out of the five test runs. The corresponding average of the first success (reward is zero for the first time) is illustrated in column $\overline{x}_F$ and the standard deviation is illustrated in column $\sigma_F$.

**Table 3.** Comparison of existing algorithms.

| Algorithm | $\overline{x}_S$ | $\sigma_S$ | $min_S$ | $max_S$ | $\overline{x}_F$ | $\sigma_F$ |
|---|---|---|---|---|---|---|
| NSGA-II | 104.2 | 15.37 | 84 | 126 | 21.0 | 13.8 |
| DDPG | 0.0 | - | 0 | 0 | - | - |
| PPO | 1.2 | 1.30 | 0 | 3 | 363.0 | 261.6 |
| A2C | 2.0 | 1.87 | 0 | 5 | 332.8 | 274.3 |
| SAC | 28.8 | 4.15 | 22 | 32 | 46.4 | 31.1 |

From the results of Table 3 it is observable that the drawbacks of Section 1 are proved and that the GA outperforms the RL algorithms. The standard deviation indicates that the results are reproducible. The only RL algorithm with some promising results for this single step problem is the SAC algorithm. The fact that one test run contained 1000 iterations only is a probable cause of the lack of performance of the RL algorithms since RL needs many iterations until promising results are available (and 1000 might be not enough) [37,41].

Therefore, the application of RL algorithms might be not appropriate for the implementation in non-simulated time-consuming development processes using hardware suffering from wear.

In contrast to the RL algorithms the NSGA-II algorithm achieves promising results shown in Figure 7:

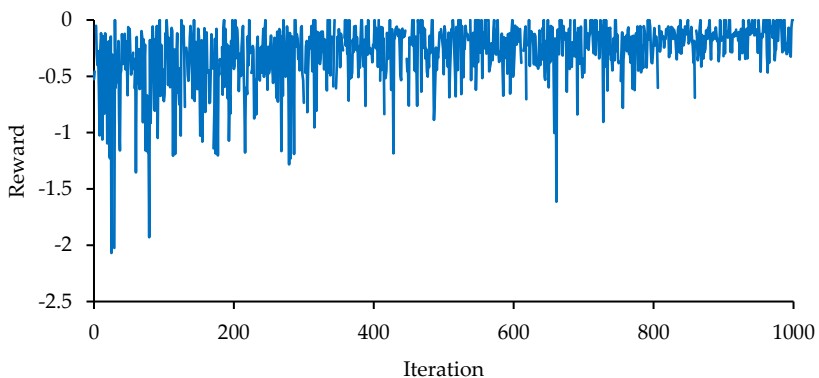

**Figure 7.** Results of an NSGA–II test run in the software in the loop environment.

As mentioned in Section 2.3, the reward does not influence the behavior of the NSGA-II algorithm (or the behavior of any GA in contrast to RL algorithms) but it is used as an indication of the quality of an action.

Due to the better performance compared to RL algorithms, the NSGA-II algorithm will be further tested within a test vehicle (Section 5.3), although there is the possibility of a worse performance with an increasing number of calibration parameters. Additionally, as shown in Figure 7, the NSGA-II algorithm produces a noisy reward behavior, with many outliers through the entire test run.

## 4. Target State Optimization

The illustrated drawbacks of the existing self-learning algorithms of Section 3 led to the new TSO approach proposed in [1]. By combining the advantages of RL and supervised learning, the TSO algorithm is a hybrid solution of these to machine learning methods.

Similar to RL, the TSO algorithm interacts with an environment. The environment is influenced by the action taken by the TSO agent and responds back with a state to the TSO agent. To illustrate the interaction of the actions with the environment and the state as the response of the environment, the following structure in Figure 8 represents the q-learning scheme according to [33]:

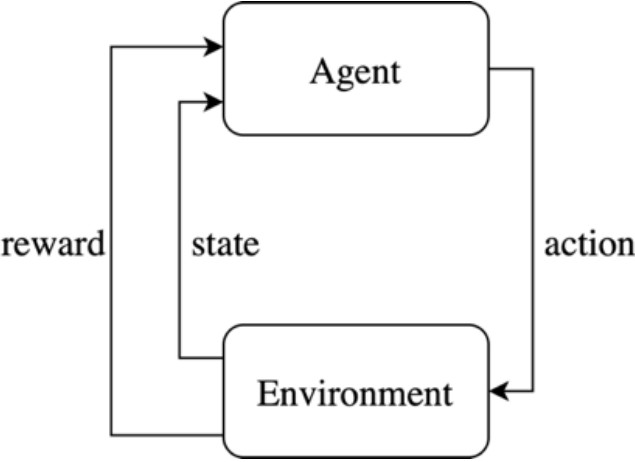

**Figure 8.** Q-learning structure according to [33].

Similar to q-learning, the agent is a neural network, which is choosing the actions. It is trained during optimization based on the dataset, which is created in parallel (in contrast a neural network is trained and optimized based on an existing dataset in supervised learning). Different to RL, the reward is not used to optimize the actions. The reward is only an indication for the quality of the action (like explained in Section 3 for genetic algorithms).

### 4.1. Generation of the Action

Optimization algorithms like GA [58] but also RL [59] algorithms suffer from the trade-off between exploration and exploitation. For finding good solutions an optimization algorithm tries to explore the state space. If a good solution has been found one option is to exploit the state space and maybe find similar or better solutions. This can lead to being stuck in a local minimum since new solutions are not being explored [58]. To ensure a trade-off between exploration and exploitation Mnih [33,60] considered the $\varepsilon$-Greedy approach for Deep Q-Learning to overcome this issue [60]:

$$\varepsilon(i) = \max\left(\varepsilon_{dec}{}^i, \; \varepsilon_{min}\right) \tag{12}$$

With this approach $\varepsilon$ is decreased in each iteration $i$ with $\varepsilon_{dec}$ until $\varepsilon_{min}$ is reached. Figure 9 illustrates the progress of epsilon for different values:

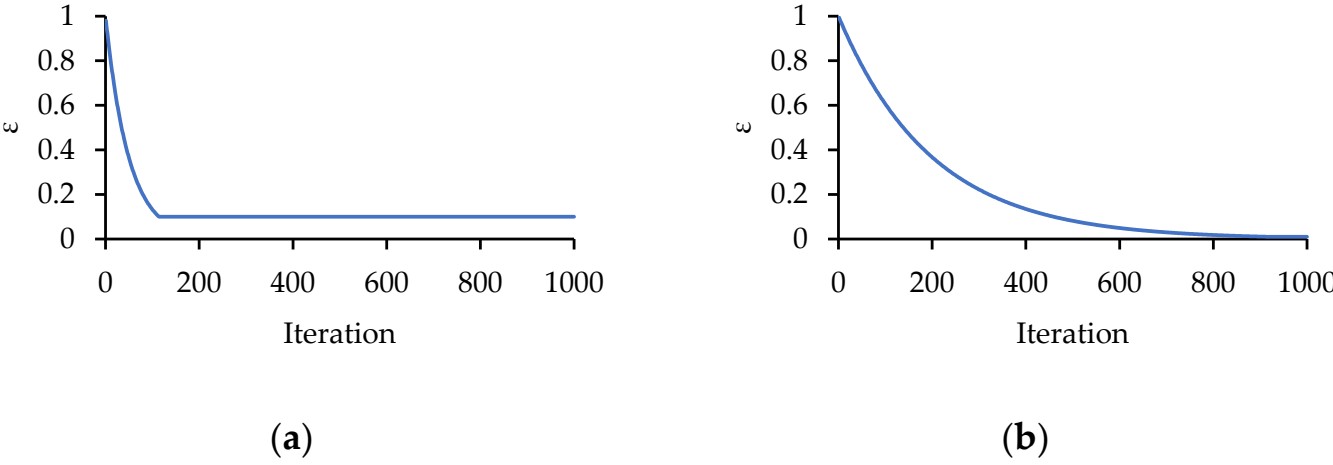

**(a)**                                                    **(b)**

**Figure 9.** Progress of $\varepsilon$ with (**a**) $\varepsilon_{dec}$ = 0.98 and $\varepsilon_{min}$ = 0.1, (**b**) $\varepsilon_{dec}$ = 0.995 and $\varepsilon_{min}$ = 0.01.

In parallel a number $r$ between 0 and 1 is generated randomly in each iteration to determine the action type. An action is either generated randomly if the random number is lower than $\varepsilon(i)$ or otherwise by the neural network. Since $\varepsilon$ is decreasing over time the first actions are primarily created randomly to explore the state space while latter actions are created by the model with exploitation in focus.

If an action is requested from the neural network model the Deep Reinforcement Learning algorithm described in Mnih [33] tries to predict an action, which maximizes the reward depending on the current state of the environment. Since the TSO algorithm is not designed to optimize combinatorial problems, the action is not dependent on a previous state. Instead, one action is passed to the environment, the environment responds with a state and the next optimization step is independent from prior states. After a certain number of iterations (*it*), the neural network learns from prior state-action pairs and hence it maps the states of the environment to the actions which led to these states. Through the random actions in early stages the assumption is that a wide state space is explored and therefore the system behavior of the environment is learnt by the neural network. Further, the optimization objectives (if an action is requested from the neural network) are passed to the neural network as a target state to predict an action. With the prior assumptions, the expectation is that the action influences the environment to a state which should be equal to the target state. If the state from the environment is not equal to the target state, the

neural network needs to be optimized or more random actions are required. If the resulting state is equal to the target state, the optimization is successful, and the reward is equal to zero. This process is illustrated in Figure 10:

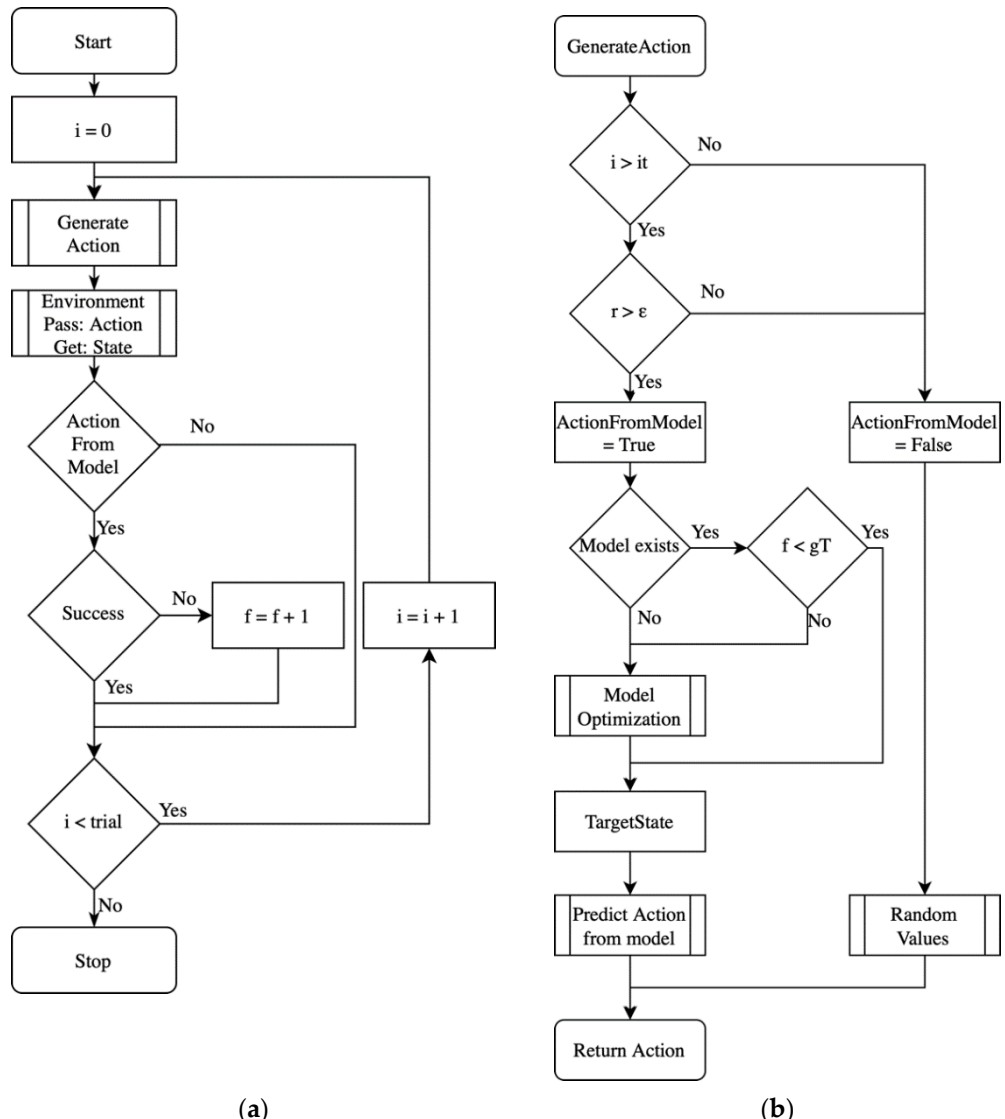

**Figure 10.** Flowchart of the TSO algorithm: (**a**) main loop, (**b**) Generate Action function.

In Figure 10a the main loop of the TSO algorithm is illustrated. It is observable that the fail count ($f$) is incremented if an action is taken by the model, which have not led to a successful result. If this fail count exceeds the threshold $gT$ (Figure 10b) or if the model is not existing, the model has to be created or optimized.

### 4.2. Model Optimization and Hyperparameter-Tuning

There are two different kind of parameters in machine learning models: model parameters (updated during learning) and hyperparameters (which cannot be learned from the data) [61]. Before starting the training process the hyperparameters have to be defined [62], which have a major influence on the accuracy of the machine learning model [63]. Therefore, it is important to adapt the hyperparameters of the machine learning model to a dataset to optimize the behavior of the model for the corresponding domain [61].

In this study, the number of layers and the number of neurons per layer (model design hyperparameters [64]) as well as the learning rate as another important hyperparameter, are optimized [65]. Choosing the learning rate is driven by the trade-off of having an accelerated

learning, but the drawback of maybe not converging (with large learning rates) and more accurate predictions but increased computation times (with small learning rates) [61].

The three hyperparameters are, as mentioned earlier, usually adapted to a dataset the model should be trained on. Since RL algorithms as well as the TSO algorithm start its optimization without having a dataset (the dataset in these algorithms is growing with the optimization progress) the optimization of the hyperparameters is different to SL where parameters are adjusted to an existing dataset. In RL algorithms the hyperparameters are usually not adjusted to the dataset during optimization so misleading initial hyperparameters can result in a lack of accuracy of the underlying neural network and therefore in an unsatisfying performance.

As described in Section 4.1, this drawback is tackled, by collecting data with randomly generated actions for a certain number of iterations ($iT$). After these iterations, and when also the $\varepsilon$-greedy approach desires the first action to be generated by the model, the hyperparameter optimization takes place to generate the neural network model (the model does not exist so far). The collected state-action pairs are used for the hyperparameter tuning, which is performed with the Bayesian optimization based on Gaussian Processes. The approach is used due to its better performance compared to other hyperparameter-tuning methods (e.g., grid-search, random-search, etc.) [66].

If the predictions of the neural network are still not correct after the hyperparameter-tuning the neural network is optimized again.

For this purpose, the fail counter ($f$), the gradient threshold ($gT$), and the tuning threshold ($tT$), are introduced. If the environmental state is not equal to the target state, $f$ is incremented. After a certain number of failed iterations ($f > gT$) the existing neural network is re-trained with the new collected training data. If the model still fails ($f > tT$), the hyperparameter-tuning is repeated and a new model $n$ is created. The optimization of the hyperparameters is only performed $m$ times (the number of hyperparameter-optimizations is limited due to computation reasons). Additionally, each new created model is compared with the ones which have already been created. This optimization process is visualized in Figure 11.

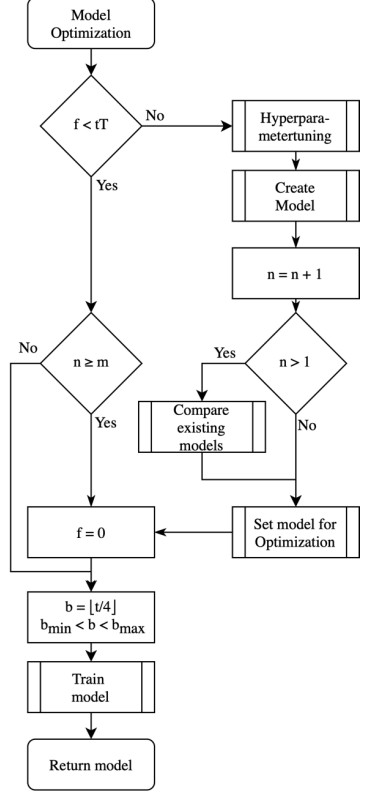

**Figure 11.** Flowchart of the model optimization.

### 4.3. Brief Introduction into Neural Networks

A neural network basically is a function estimator, which computes the relationship between the input of a neural network and its output [67]. During hyperparameter-tuning (Section 4.2 the number of units (neurons) of each layer and the number of layers is determined. The input of a single unit $a$ is determined with a set of input variables $d$ which is weighted and summarized [68]:

$$a = \sum_{i=1}^{d} w_i \cdot x_i + w_0 \tag{13}$$

The output of a unit is determined by processing $a$ within a non-linear activation function $g$:

$$z = g(a) \tag{14}$$

For multiple units with the same inputs (single layer neural network) the representation changes to:

$$z_j = g\left( \sum_{i=0}^{d} w_{ji} \cdot x_i \right), \tag{15}$$

for $x_0 = 1$. For multiple layers the output of next layer $(j + 1)$ is based on the output of the former layer represented in:

$$z_{j+1} = g\left( \sum_{j=0}^{m} w_{(j+1)j} \cdot z_j \right) \tag{16}$$

So, for a network with two layers the notation of $z_{j+1}$ is represented in:

$$\hat{y} = z_{j+1} = g\left( \sum_{j=0}^{m} w_{(j+1)j} \cdot g\left( \sum_{i=0}^{d} w_{ji} \cdot x_i \right) \right) \tag{17}$$

To correlate the input values $(x)$ to the output values $(y)$ the weights are determined with existing data of input and output values. This process is called training [68].

The training is performed by comparing the predicted value of the neural network $\hat{y}$ from an input $x$ with its true value $y$. Since values are passed through the network the step is called forward propagation. To compare the predicted and the true value an error function is introduced. A common error function is the mean squared error:

$$E = \frac{1}{2} \sum_{q=1}^{n} \left( \hat{y}(x_q, w) - y_q \right)^2 \tag{18}$$

The error of different values of the dataset are summed up. The error is minimized by adapting the weights during backpropagation.

By inserting Equation (17) into Equation (18) the error becomes a function of the weights and hence if the weights are changing the error changes as well. Additionally, the function can be differentiated. During backpropagation the derivative of the function is determined, and the error function is minimized by making fixed steps (learning rate) into the direction of the negative gradients (gradient descent) [68]. The determination of the learning rate is also part of the hyperparameter-tuning of Section 4.2.

### 4.4. Relevance of the Activation Function

As shown in Section 4.3, the activation function influences the determination of the weight vector and the training effort since it is part of the error function. The activation function is also a hyperparameter, but it is not part of the hyperparameter-tuning in TSO of Section 4.2 to reduce the computation time. Therefore, this section is introduced to investigate the influence of three common activation functions on the optimization effort:

the linear identity activation function, the rectified linear activation function (ReLU), and the logistic sigmoid activation function [69]. These activation functions are represented in Figure 12:

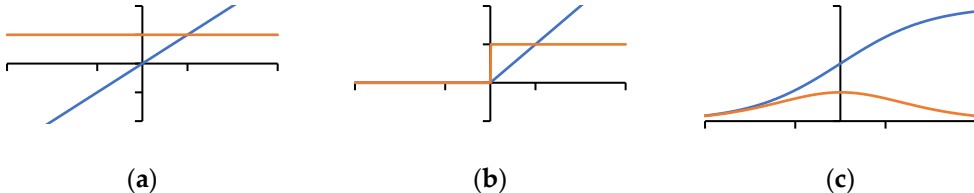

$(\mathbf{a})$ $(\mathbf{b})$ $(\mathbf{c})$

**Figure 12.** (**a**) linear, (**b**) ReLU, and (**c**) logistic sigmoid activation function (blue) and its derivative (orange).

The linear activation function is represented as:

$$g(x) = x \tag{19}$$

According to Bishop [68] using the linear activation function has the drawback of reducing the neural network to single-layer neural network and hence a simple matrix multiplication, which comes with limited computational capabilities. Therefore, it is mandatory to have more than one layer (deep neural networks) for more sophisticated tasks [68]. This requires non-linear activation functions like the ReLU function [70]:

$$g(x) = \max(0, x) \tag{20}$$

or the logistic sigmoid activation function [68]:

$$g(x) = \frac{1}{1 + e^{-x}} \tag{21}$$

However, the growing number of layers lead to a greater amount of weights and therefore, the weight estimation becomes more computationally expensive [70]. According to Schmidt-Hieber [70] the rectified linear unit (ReLU) has an computational advantage compared to sigmoidal functions like the logistic sigmoid function. The influence of the selection of the activation function is shown in the results in Section 5.

*4.5. Batch Size*

The batch size is another hyperparameter, which determines the number of training samples used to update the neural network model [71]. If the batch size is too large it can result in bad generalization and lead to false predictions. In contrast a too small batch size results in a better quality but an enlarged computation time [72]. The batch size is usually kept constant but due to the fact that the dataset is increasing with the optimization progress this approach does not seem appropriate for TSO. Until the dataset has the same size as the batch size the entire dataset is learned by the neural network. If the batch size is kept constant a large batch size could lead to wrong predictions over a longer time period, and a small batch size could lead to an increased computation time with an increasing dataset. Therefore, the batch size is in TSO dependent on the dataset size:

$$b = \frac{n}{4} \tag{22}$$

$$b_{min} < b < b_{max} \tag{23}$$

The batch size is determined dynamically according to the optimization progress with $n$ as the size of the dataset. $b_{min}$ and $b_{max}$ are the minimum and maximum values of the batch size.

## 5. Results

### 5.1. Software in the Loop

To compare the TSO algorithm with the best benchmark algorithms of Section 3, the TSO algorithm is applied on the optimization problem of Section 2 within the software in the loop environment.

Table 4 is introduced to illustrate the results. Again, as in Table 3, the columns illustrate the average values ($\bar{x}_S$ and $\bar{x}_F$), the standard deviation ($\sigma_S$ and $\sigma_F$), and the result range ($min_S$ and $max_S$) of the "successful iterations" and the "first success" over the five test runs.

**Table 4.** Comparison of the TSO algorithm with the benchmark algorithms.

| Algorithm | $\bar{x}_S$ | $\sigma_S$ | $min_S$ | $max_S$ | $\bar{x}_F$ | $\sigma_F$ |
|---|---|---|---|---|---|---|
| NSGA-II | 104.2 | 15.4 | 84 | 126 | 21.0 | 13.8 |
| SAC | 28.8 | 4.2 | 22 | 32 | 46.4 | 31.1 |
| TSO (sigmoidal) | 7.8 | 5.5 | 2 | 16 | 221.6 | 332.0 |
| TSO (ReLU) | 250.0 | 160.4 | 36 | 442 | 20.6 | 11.3 |

It is observable that the activation function (Section 4.4) has a significant impact on the results. While the TSO algorithm with the ReLU activation function outperforms the GA and the best tested RL algorithm (with regard on successful iterations), the TSO algorithm with the sigmoidal activation function lacks in performance which is observable in Figure 13.

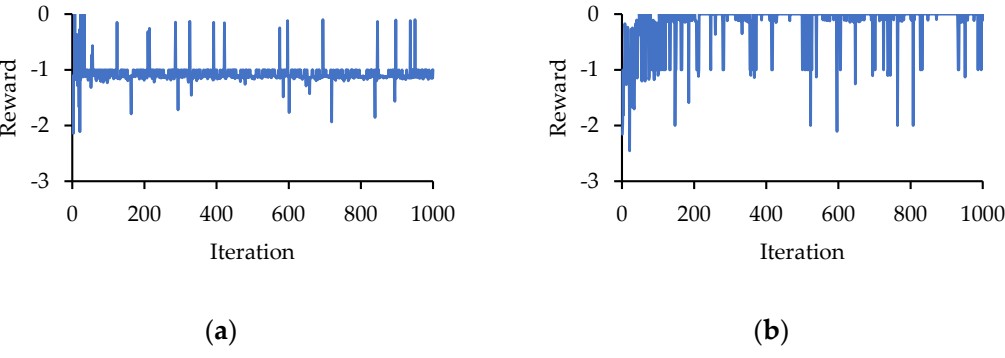

(**a**)            (**b**)

**Figure 13.** Comparison of the influence of the activation functions of the TSO algorithm on the re–sults: (**a**) sigmoidal activation function, (**b**) ReLU activation function.

Although the standard deviation is very high compared to the other algorithms (one outlier with only 36 successful iterations, which is significantly low compared to the other test runs), the results show that for a fast and reliable success the ReLU activation function has to be used for the TSO algorithm. It is also noticeable that the first successful iteration generated is almost the same for the NSGA-II and the TSO algorithm with the ReLU activation function.

Different to the activation function, the hyperparameters explained in Section 4.2 are adjusted during the optimization. The hyperparameter-tuning is carried out three times. The progress of the hyperparameter tuning for the TSO algorithm with the ReLU activation function is illustrated in Table 5 for two different test runs.

Due to the success within the software in the loop environment, the algorithms NSGA-II and TSO are further tested in different test vehicles. Within the test vehicles the last combination of the hyperparameters (layers = 4, neurons = 425, learning rate = 0.0012) is tested to optimize the calibration parameters.

**Table 5.** Hyperparameter propagation.

| Test Run/Iteration | Layer | Neurons | Learning Rate |
| --- | --- | --- | --- |
| 7 | 1 | 512 | 0.0237 |
| 16 | 1 | 25 | 0.0545 |
| 23 | 1 | 512 | 0.0237 |
| 6 | 6 | 130 | 0.0247 |
| 16 | 1 | 180 | 0.0414 |
| 20 | 4 | 425 | 0.0012 |

*5.2. Robustness*

According to Kitano [73], robustness is the ability of a system to function under perturbations and hence under changing conditions. For the TSO algorithm, it is mandatory to maintain quasi-stationary conditions during optimization to find optimal solutions. Improving the behavior of a vehicle launch e.g., under flat conditions, shall not be performed at test facilities with high slopes. However, it is difficult to find stationary environmental conditions outside a simulation environment since test tracks are usually influenced by the environment. Changing conditions can lead to changing driving resistances (e.g., small slopes) and hence the system behaves different compared to optimal conditions. Therefore, before testing the algorithm in a real-world vehicle, the robustness of the vehicle is tested within the software in the loop environment by changing the slope in the simulation spontaneously to simulate the conditions of real roads. A quasi-flat road is simulated by varying the slope arbitrary between −1% and 1% as well as between −2% and 2%.

To test the behavior the optimizations are repeated four times in the software in the loop environment with 100 iterations in each test run. The results are shown in Table 6.

**Table 6.** Successful iterations of the robustness test with the TSO algorithm.

| Test Run | ±1% | ±2% |
| --- | --- | --- |
| 1 | 20 | 21 |
| 2 | 78 | 33 |
| 3 | 17 | 74 |
| 4 | 7 | 0 |
| Average | 30.5 | 32 |

It is observable that the results vary strongly in each test run. While one test run has 78 successful iterations another one does not have any successful iteration. In average 30.5 test runs have been successful for ±1% and 32 for ±2% road gradient.

Although the results are varying between the test runs, it also indicates a potential for the application in a test vehicle.

*5.3. Test Vehicle*

Within the test vehicle the NSGA-II and the TSO algorithm are validated again with equal objective values which are illustrated in Table 7:

**Table 7.** Optimization targets.

| Objective | Value | Unit |
| --- | --- | --- |
| Engine Speed Drop | 0.00 | rpm |
| Acceleration Peak | 3.25 | $m/s^2$ |
| Clutch Torque Minima | 0.00 | - |
| Reaction Time | 0.50 | s |

The test vehicle is a vehicle with a hybrid dual clutch transmission (HDT) and a three-cylinder Otto engine. The HDT is basically a conventional dual clutch transmission with an electric motor applied. The electric motor is able to transmit torque if the second of the

two clutches is closed [74]. This enables the powertrain to reduce the fuel consumption by 14.5% compared to powertrains with conventional DCT [75]. Due to confidential reasons the powertrain configuration cannot be explained in detail.

For the optimization of the driving behavior during launch the number of iterations for optimization in a test vehicle is tried to be kept as low as possible to reduce development costs. Therefore, two different hyperparameter configurations are applied to test the NSGA-II algorithm. The first configuration is performed with 20 iterations containing five generations and a population size of four (an explanation of these parameters can be found in Section 1). The second configuration is performed with 60 iterations and also with five generations but with a population size of 12. The results are illustrated in Figure 14:

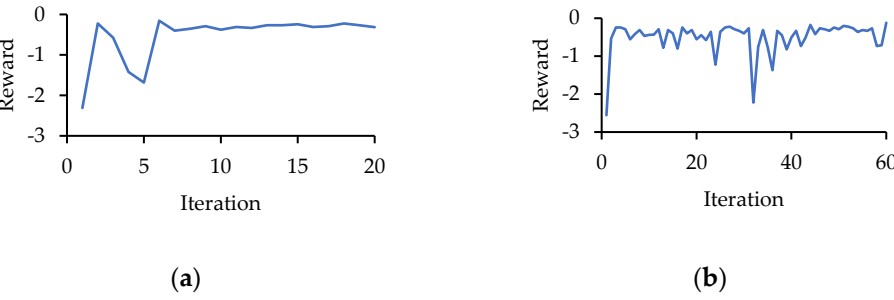

(**a**)                                                    (**b**)

**Figure 14.** NSGA–II results tested in a vehicle with an HDT and a three-cylinder Otto engine: (**a**) generations: 5, population: 4; (**b**) generations: 5, population: 12.

It is observable that the reward is always lower than zero and hence the optimization target has not been reached with neither algorithm configuration.

To compare the results of the TSO algorithm with those of the NSGA-II algorithm, a lower number of iterations are also applied during optimization. However, the optimization is performed in the same vehicle under equal environmental conditions. The optimal parameters for a successful launch are tried to be found within 50 iterations. To reduce the computation time hyperparameter-tuning is neglected in the vehicle tests. The neural network is initialized with the last hyperparameters identified in the software in the loop environment (number of layers = 4, neurons per layer = 425, learning rate = 0.0012). The results are shown in Figure 15:

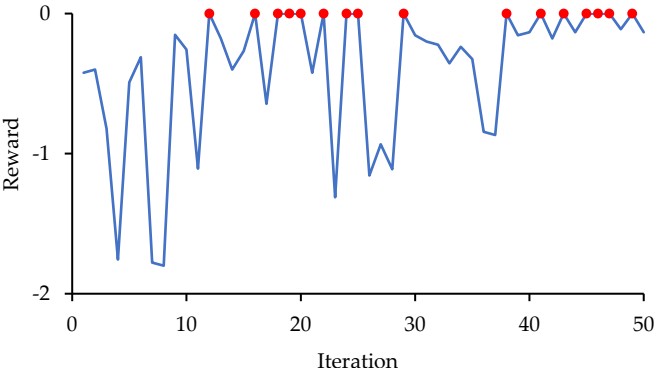

**Figure 15.** Results of the TSO algorithm tested in a test vehicle with an HDT and a three–cylinder Otto engine.

Out of 50 iterations, 16 have been successful with the first successful result was at iteration 11. Hence, the TSO algorithm again outperforms the NSGA-II algorithm.

The difference of a successful (reward = 0) and a failed launch is illustrated in Figure 16.

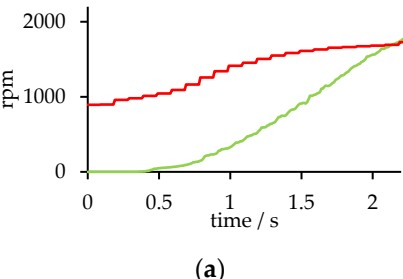 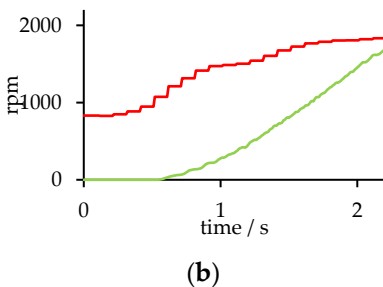

(a) (b)

**Figure 16.** (**a**) Successful launch (TSO: iteration 11 of Figure 15), (**b**) failed launch (NSGA-2: iteration 0 of Figure 14a); Engine Speed (red); Input-shaft speed (green).

The clutch torques of these launches are illustrated accordingly in Figure 17.

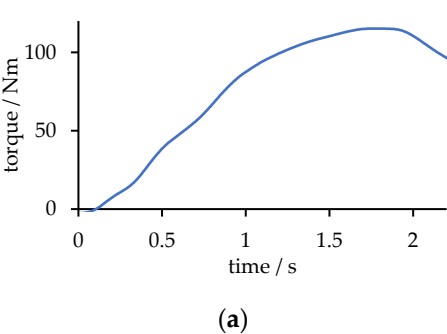 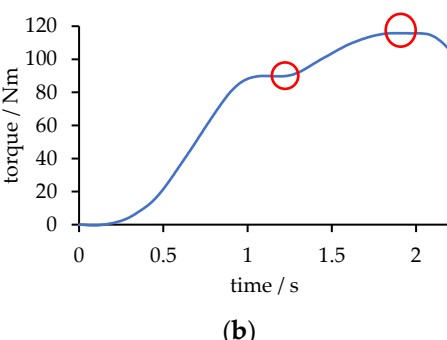

(a) (b)

**Figure 17.** Clutch torques of the different launches accordingly to Figure 16: (**a**), desired clutch torque behavior (**b**) clutch torque with local minima.

In Figure 16a, no drop of the engine speed, and in Figure 17a no local minima of the clutch torque, is observable. Additionally, the measured maximum acceleration is 3.26 m/s$^2$ and the reaction time is 0.45 s, which is within the tolerance and hence lead to a reward of zero. In Figure 16b, an inconsistency of the engine speed is observable, but the engine does not drop and hence it does not influence the evaluation negatively. Figure 17b illustrates two local minima of the clutch torque (marked red). The minimum at timestamp 1.18 s is clearly observable compared to the one at timestamp 1.96 s. Additionally, the measured reaction time of 0.59 s is out of the tolerance which upper threshold would be at 0.55 s. In contrast, the maximum acceleration of 3.24 m/s$^2$ is within the tolerance. The reward is $-2.31$ and hence the launch failed.

To illustrate the ability to generalize, the algorithm is also applied to another vehicle with a three-cylinder Otto engine, but this time equipped with a conventional DCT. However, first for comparison, again the NSGA-II algorithm is used to optimize the driving behavior.

In contrast to Figure 14 in Figure 18b one successful launch has been determined which could not be reproduced in the optimization progress.

Figure 19 illustrates that the TSO algorithm still outperforms the NSGA-II algorithm, but it is also observable that the optimization objectives should have been adjusted to the different powertrain configuration since a successful launch has not been achieved as often as in Figure 15.

To verify the ability of the algorithm to solve the optimization problem in any vehicle, the algorithm is tested in another vehicle. This time, only the TSO algorithm is tested in a vehicle with a four-cylinder diesel engine and a 7-speed HDT. Again, the target state of Table 7 is used with a reference acceleration of 3.25 m/s$^2$ and a reference reaction time of 0.5 s.

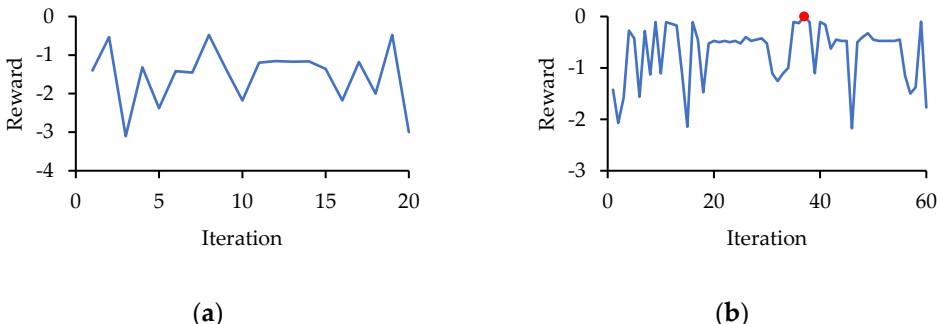

(**a**) (**b**)

**Figure 18.** NSGA–II results tested in a test vehicle with a conventional DCT and a three–cylinder Otto engine: (**a**) generations: 5, population: 4; (**b**) generations: 5, population: 12.

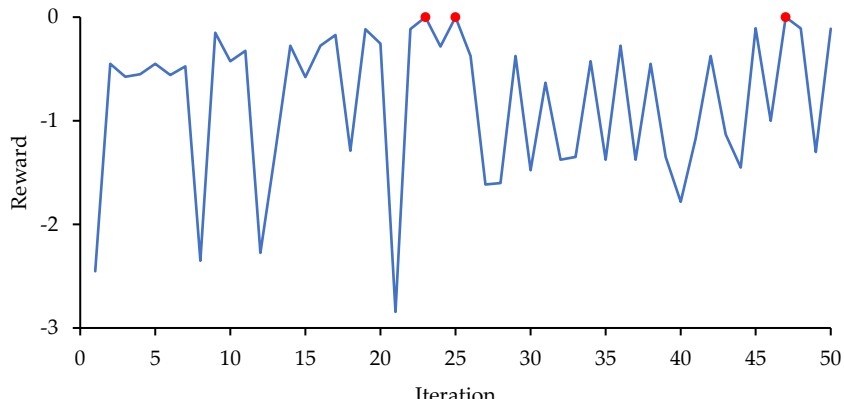

**Figure 19.** Results of the TSO algorithm tested in a test vehicle with a conventional DCT and a three–cylinder Otto engine.

Figure 20 illustrates that the optimization targets have not been reached by the algorithm. Using a four-cylinder diesel engine comes with the side-effect of having a different behavior of the engine torque compared to Otto-engines, as well as having comparably more power by having a fourth cylinder (the accelerator pedal positions has been set to the same value as in the other tests). The differences in the physical behavior of the other engine configuration leads to the fact that the former targets cannot be reached with this engine-transmission combination. Therefore, different optimization targets are tested within vehicle with the diesel engine. The results are illustrated in Table 8:

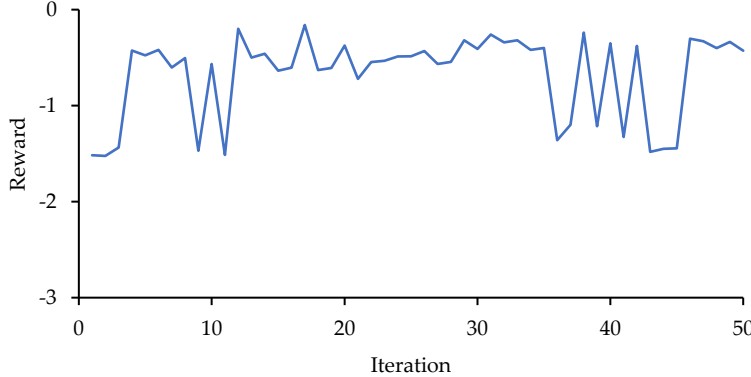

**Figure 20.** Results of the TSO algorithm tested in a test vehicle with an HDT and a four–cylinder diesel engine (Acceleration target: 3.25 m/s$^2$, reaction time target: 0.5 s).

**Table 8.** Comparison of the TSO algorithm with different optimization targets within the vehicle with a four-cylinder diesel-engine and an HDT.

| Acceleration | Reaction Time | Successful Iterations | First Success |
|---|---|---|---|
| $3.25 \ \text{m/s}^2$ | 0.5 s | 0 | - |
| $3.5 \ \text{m/s}^2$ | 0.3 s | 16 | 13 |
| $3.5 \ \text{m/s}^2$ | 0.4 s | 11 | 4 |
| $4 \ \text{m/s}^2$ | 0.3 s | 1 | 13 |

Table 8 illustrates that an acceleration target of $3.5 \ \text{m/s}^2$ and a reaction time target of 0.3 s for this engine-transmission combination and this load come with the greatest number of successful launches. A higher reaction time with the same acceleration can also be reached by the system behavior, but a too high or too low chosen acceleration target lead to poor results.

The different tests illustrate that the TSO algorithm is in general able to optimize the behavior of a vehicle launch by avoiding discomfort (torque and engine speed inconsistencies), but obviously only if the acceleration and reaction time targets (customer requirements) are reachable.

## 6. Discussion

In the scope of the TCU parameter optimization problem, the TSO algorithm found the desired optimum (derived from Section 2) faster and hence outperformed common RL algorithms and the GA NSGA-II. Table 4 illustrates the better performance of the TSO algorithm compared to the other algorithms. It is observable that the TSO algorithm achieved almost nine times more successful results compared to the best tested RL algorithm (SAC) and more than twice as many successful iterations compared to the GA NSGA-II. The results of the TSO algorithm were even better in the test vehicle. The parameter optimization of the vehicle with the conventional DCT and a three-cylinder Otto engine with the NSGA-II algorithm (Figure 18b) only had one successful result. The other tests with the NSGA-II algorithm have not been successful. In contrast, the TSO algorithm achieved more successful results in the tests with the 3-cylinder Otto engines with the HDT (Figure 15) and in the vehicle with the conventional DCT (Figure 19).

However, the performance of the TSO algorithms is varying between these two tests, which indicates that some domain knowledge is still mandatory to choose the right optimization targets. Therefore, the optimization with the third test vehicle (4-cylinder diesel engine with an HDT applied) has been carried out with different combinations of the optimization objectives to identify the system boundaries. It is observable that some objectives cannot be reached in every tested vehicle. This problem is obviously also existing for the other algorithms but a workaround to shorten the optimization time would be beneficial for further studies.

An approach for further studies could be the adjustment of the reference acceleration (Section 2.1.1) and the reaction time (Section 2.1.2) dynamically depending on the system behavior (e.g., striving for the minimization of the reaction time while meeting the other conditions would be beneficial for the driving behavior).

Furthermore, the robustness tests of Section 5.2 illustrated that further investigations need to be carried out although the promising results in the different test vehicles (illustrated in Section 5.3) indicated a good generalization. The application of the TSO algorithm for other optimization problems is pending.

## 7. Conclusions

The time-consuming calibration process of vehicle control units has so far tried to be automized with different methods. Despite the proposed methods the calibration process is usually still driven by calibration engineers. A new promising approach is illustrated in this study with the TSO algorithm as a hybrid of RL and SL. The advantage of the TSO

algorithm is that it, unlike RL approaches, does not try to maximize a reward, instead a target state is approached. Besides the actual optimization, the algorithm also optimizes the hyperparameters of the underlying neural network (compared to RL approaches) with Bayesian optimization based on Gaussian Processes, and hence fits the network to the optimization problem (Section 4.2). Additionally, the activation function of the TSO algorithm has to be chosen carefully since it strongly influences the performance of the algorithm. The investigations shown in Section 4.4 illustrate that the ReLU function turned out to be beneficial for this problem.

To automize the optimization it is further necessary to transfer subjective feelings into objective measurements. Therefore, the TSO algorithm is tested regarding prior subject studies aiming at the improvement of the drivability. Hence, the influence of the objectives is investigated, and their EDTs are identified with test subject studies. The launch behavior can be evaluated empirically with these outcomes in relation to sportiness, comfort, jerkiness, and agility. In order to ensure a comfortable launch, the maximum acceleration and the jerk caused by clutch torque should be controlled at an acceptable level.

In the scope of the parameter optimization problem for TCUs, the TSO algorithm constitutes a new benchmark compared to the other tested algorithms with its promising results. The study proves that the fast converging (not only in simulations but also in real world applications) enables the algorithm to be implemented into the existing development process and simplifies the work of calibration engineers. Therefore, the TSO algorithm increases the efficiency of the calibration process and possibly decreases costs of vehicle manufacturers and suppliers by applying computer science methods in ongoing development processes in the automotive industry.

**Author Contributions:** Conceptualization, M.S. and P.H.; methodology, M.S. and P.H; software, M.S.; validation, M.S. and P.H.; formal analysis, M.S.; investigation, M.S.; resources, M.S.; data curation, M.S.; writing—original draft preparation, M.S. and P.H; writing—review and editing, S.R.; visualization, M.S. and P.H; supervision, S.R.; project administration, S.R. All authors have read and agreed to the published version of the manuscript.

**Funding:** We acknowledge support by the Deutsche Forschungs- gemeinschaft (DFG—German Research Foundation) and the Open Access Publishing Fund of Technical University of Darmstadt.

**Institutional Review Board Statement:** The study was conducted according to the guidelines of the Declaration of Helsinki and approved by the Ethics Committee of Technical University of Darmstadt (protocol code: EK 51/2021, date of approval: 23 November 2021).

**Informed Consent Statement:** Informed consent was obtained from all subjects involved in the study.

**Data Availability Statement:** Not applicable.

**Conflicts of Interest:** The authors declare no conflict of interest.

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
