# Peer review of "Target State Optimization: Drivability Improvement for Vehicles with Dual Clutch Transmissions"

_applsci, doi:10.3390/app122010283_

Round 1
Reviewer 1 Report
Dear authors,
I have read the manuscript carefully. The manuscript is overall well structured and defined. However, I have several suggestions, which in my opinion will improve the quality of the paper.
In order to better the define aim and problem of the paper, maybe is a good idea to change the end of the introduction with hypotheses which will be proved or refuted trough the paper. Also, insted of hipotheses, can be defined questions on which will be answered trough the paper.
Also, the methodology of investigation is quite in incomprehensible, so this should be a little better described in the paper.
Also, some Figures are low quality, so this also shod be modified, as well as, it seems to me that the font on some Figures is different than the font which was used in the paper.
The conclusion is written quite descriptive, without the focus on the main findings from the paper, so the conclusion should be rewritten.
Reviewer 2 Report
1. Manuscripts with reference value. Please revise the following minor errors
2. The objectives on the fourth page seems to be incomplete, or there is an extra "and" at the end of the sentence
3. section 2.1.1 …. on the evaluation criteria: of sportiness…. The colon seems in the wrong place.
4.The discussion in [7] seems to be the basis for the construction of this paper, otherwise it is recommended to conduct a comprehensive exploration of [7], and conduct in-depth research on the usability and improvement of its content.
5.The figure is suggested to be cited, it should also be marked with the reference number after the description. As in FIG.2, the subgraph seems less understandable to the reader.
6.From the description, it is not eazy to confirm whether Fig. 3 is the data of Reference 7 or the result of this research. In addition, the meaning represented by the coordinates should still be marked.
7.The red marker in Fig. 5 seems to have the same phenomenon at 0.5 seconds. What standard is used to judge the "discomfort". The same problem as Fig7. Compare with NSGA-II algorithm.
8.“Due to the success within the software in the loop environment the algorithms NSGA-II and TSO are further tested in different test vehicles.” Does it refer to the same vehicle type or different vehicle types, and how to compare the results of different vehicle types?
Round 2
Reviewer 1 Report
Authors improved their manuscript, so I suggest paper acceptation in its current form.